# Peptides of a Feather: How Computation Is Taking Peptide Therapeutics under Its Wing

**DOI:** 10.3390/genes14061194

**Published:** 2023-05-29

**Authors:** Thomas David Daniel Kazmirchuk, Calvin Bradbury-Jost, Taylor Ann Withey, Tadesse Gessese, Taha Azad, Bahram Samanfar, Frank Dehne, Ashkan Golshani

**Affiliations:** 1Department of Biology, and the Ottawa Institute of Systems Biology (OISB), Carleton University, Ottawa, ON K1S 5B6, Canada; 2Department of Microbiology and Infectious Diseases, Université de Sherbrooke, Sherbrooke, QC J1E 4K8, Canada; 3Centre de Recherche du Centre Hospitalier Universitaire de Sherbrooke (CHUS), Sherbrooke, QC J1H 5N4, Canada; 4Agriculture and Agri-Food Canada, Ottawa Research and Development Centre (ORDC), Ottawa, ON K1A 0C6, Canada; 5School of Computer Science, Carleton University, Ottawa, ON K1S 5B6, Canada

**Keywords:** peptide drugs, peptide design, computational biology, artificial intelligence, protein–protein interaction, InSiPS

## Abstract

Leveraging computation in the development of peptide therapeutics has garnered increasing recognition as a valuable tool to generate novel therapeutics for disease-related targets. To this end, computation has transformed the field of peptide design through identifying novel therapeutics that exhibit enhanced pharmacokinetic properties and reduced toxicity. The process of *in-silico* peptide design involves the application of molecular docking, molecular dynamics simulations, and machine learning algorithms. Three primary approaches for peptide therapeutic design including structural-based, protein mimicry, and short motif design have been predominantly adopted. Despite the ongoing progress made in this field, there are still significant challenges pertaining to peptide design including: enhancing the accuracy of computational methods; improving the success rate of preclinical and clinical trials; and developing better strategies to predict pharmacokinetics and toxicity. In this review, we discuss past and present research pertaining to the design and development of *in-silico* peptide therapeutics in addition to highlighting the potential of computation and artificial intelligence in the future of disease therapeutics.

## 1. Background

Proteins are molecular machines that perform a diverse set of cellular processes, frequently forming complexes via protein–protein interactions (PPIs). As a result, PPIs have emerged as important targets for drug development, with peptide therapeutics designed to disrupt these interactions garnering attention due to their high specificity and low immunogenicity [1,2,3]. Most PPIs are thought to be formed as a direct result of the docking of 3D structures between proteins. However, growing evidence suggests that a notable subset of interactions is mediated by short motifs (SMs) constituting an alternative mode of PPI. SMs are involved in a variety of endogenous processes ranging from receptor signaling cascades to host-pathogen interactions [4], mediating approximately 15–40% of PPIs [5].

Chemical inhibitors (drug compounds, antibiotics, etc.) have been historically preferred for therapeutics due to their ease of synthesis and cell membrane permeability [2]. However, they generally suffer from low specificity leading to off-target interactions and side effects. Peptide therapeutics have been found to exhibit high specificity [1]. However, predicting their structure and interacting partners is challenging, impeding their general applications. To address this challenge, several commercial and research groups (i.e., DeepMind, Baker’s Lab) have employed high-throughput machine learning algorithms trained on numerous known protein structures to better study the biology of peptide therapeutics on these structures [6,7]. Although these methods have demonstrated their efficacy, they require significant computational resources and exhibit substantial complexity, posing a barrier to their widespread adoption. Recently, algorithms based on SMs that disrupt PPIs have emerged as a promising alternative [5,8,9,10]. These suggest that it may not be necessary to consider a protein drug’s structure to assess its capacity for specific interaction with a target, providing an accessible avenue for drug development, particularly for proteins with poorly understood structures and interactions.

Generating highly specific peptide therapeutics based on protein structure is computationally intense [11]. Computational methods—such as a molecular dynamics simulation—can model molecular behavior over time thereby providing insights into peptide function and interactions. Structural-based peptide design utilizes insights from protein 3D structures to systematically develop and refine peptide-based therapeutics targeting specific biological functions [12,13,14]. Meanwhile, protein mimicry entails designing proteins to imitate small molecules, including peptides. This enables critical molecular reactions and the development of drugs and therapeutics for various diseases [15]. Different protein mimicry approaches including DeNovo and structure-guided design can be used to stabilize protein structure, design drugs, and produce vaccines [15,16,17]. Furthermore, combining SMs with various specificities can lead to the creation of multi-functional peptides that can target multiple pathways or bind to multiple protein targets [18]. SMs are highly conserved across species making it possible to use peptides designed to target different organisms and reducing the efficacy gap between species when testing peptide therapies in animal models [19,20].

Peptide therapeutics generated *in-silico* using advanced algorithms and AI offer new possibilities for generating peptide therapeutics with high specificity [1]. Computation can aid in designing peptide structures to mimic protein activity, providing insight into important disease pathways, cellular localization, and potential therapeutic applications [15,16,17]. Peptide therapeutics developed through computation can therefore provide potential avenues for generating therapeutics against a broader range of drug targets [10]. Peptide design primarily involves three related technologies each of which offers a distinct perspective: structural-based, protein mimicry, and SM design (Figure 1). Here, we discuss past and present *in-silico* peptide design technologies and the future of AI in the development of peptide therapeutics.

## 2. Algorithm and AI-Assisted Structural Design

Multiple techniques are available for predicting the 3D structure of peptides based on physical principles [21]. Empirical rules are dependent on hydrogen bonding, steric hindrance, and conformational preferences of amino acid residues, and can predict the peptide’s secondary structure [22,23,24]. Statistical potentials are derived from statistical analysis of known protein structures and can evaluate the compatibility of different peptide conformations with observed protein structures whereas physics-based potentials attempt to model the energy landscape of protein folding and can predict the stability of different peptide conformations [21,25,26]. Molecular dynamics simulations utilize computational techniques to model the motion of atoms and molecules over time and provide information about the 3D structure, dynamics, and energetics of the peptide [27]. However, the accuracy of the results depends on the quality of the force field parameters, which can be calibrated to match experimental data or calculated *ab-initio*, and the simulation outcomes should be verified by experimental data whenever possible [28,29]. These methods provide essential insights into the 3D structure and dynamics of peptides, but their accuracy and computational efficiency have limitations.

Various techniques are used to predict the 3D structure of a peptide using de novo design methods [30]. Fragment assembly methods involve predicting the structure of a peptide by combining smaller fragments of known structures to build up the full structure of the peptide. *Ab-initio* modeling predicts protein structure by searching for the lowest energy conformation of the peptide backbone. Another commonly used computational method is I-TASSER, which generates full-length models by combining template-based modeling with fragment assembly simulation [31].

A de novo structural design method known as Rosetta is a molecular modeling software suite that can predict the 3D structure of a protein using a Monte Carlo search algorithm [29]. It can use various experimental data (including NMR spectroscopy and X-ray crystallography) to improve the accuracy of its predictions. Rosetta’s protein–protein docking tool can predict the structure of complexes between two or more proteins, which is useful for understanding how proteins interact with each other. This in turn is an important point to consider when designing and generating peptide therapeutics [32]. Transform-restrained (tr)-Rosetta was later leveraged to predict inter-residue contact maps for a given sequence [33]. To achieve this, a loss function was defined as the Kullback–Leibler divergence between the contact map predicted by the tr-Rosetta neural network and a background distribution. This loss function was optimized through Monte Carlo-simulated annealing, allowing for the simultaneous design of novel sequences and structures [33]. Experimental characterization confirmed that diverse folded structures were successfully designed using this model. Coarse-grained modeling is a computational technique that simplifies the representation of biomolecules (such as peptides) by reducing the number of degrees of freedom and using fewer atoms to represent the system [34,35]. In this approach, the peptide is represented as a series of beads, where each bead represents a group of atoms in the peptide backbone or side chains [35]. Coarse-grained models can be used to generate an initial model of the peptide or protein, which can then be refined using more detailed methods, such as molecular dynamics simulations or all-atom modeling.

### 2.1. Co-Evolutionary Analysis

Co-evolutionary analysis involves predicting the 3D structure of a peptide by analyzing the co-evolutionary patterns of amino acids within the peptide sequence. This method involves analyzing the patterns of sequence variation among homologous proteins or peptides [36]. If two residues are in proximity in the 3D structure, changes in one residue are likely to be accompanied by compensatory changes in the other residue to maintain the overall stability of the structure. Therefore, if two residues are co-evolving this suggests that they are in proximity in the 3D structure. While these techniques provide valuable insights into the 3D structure and dynamics of peptides and proteins, they are limited by their computational inefficiency [37].

### 2.2. Structural Alphabet

A structural alphabet method is a powerful computational approach to designing peptide structures that simplifies complex protein structures by decomposing them into small, repetitive structural building blocks known as “letters” or protein blocks [38]. The method aims to capture the essential structural features of a protein while reducing the complexity of the system, thus making it an efficient tool for designing peptides with desired properties, especially in situations where experimental data on a protein structure is limited or when dealing with large protein structures [39]. Structural alphabet methods, such as Backbone-based Rotamer Library (BbRL) and PEP-FOLD3, offer unique advantages in peptide design and can lead to the creation of novel peptides with specific backbone conformations and sidechain rotamers [40,41].

BbRL is a structural alphabet method that generates a library of backbone-dependent rotamers, which can be used to design peptides with specific backbone conformations and sidechain rotamers [40]. This method enables researchers to predict side-chain conformations with a high degree of accuracy. Meanwhile, PEP-FOLD3 is a computational framework that can predict the 3D structure of linear peptides ranging from 5 to 50 amino acids [41]. PEP-FOLD3 can perform de novo free or biased prediction and can generate native-like conformations of peptides interacting with a protein when the interaction site is known [41]. This makes PEP-FOLD3 an efficient tool for predicting the structure of linear peptides and understanding their interactions with proteins, which can be useful in drug discovery and other therapeutic applications. Methods that use structural alphabet are powerful computational tools that can help researchers design peptides with specific structural features and properties. BbRL and PEP-FOLD3 are examples of such methods that can simplify protein structures, reduce the complexity of the system, and facilitate peptide design and optimization for a range of biological applications [40,41].

### 2.3. Deep Learning and Machine Learning

AI architectures have revolutionized the field of structural biology by enabling the accurate prediction of complex peptide structures [42]. Among these methods, AlphaFold stands out as a deep learning-based approach. AlphaFold accurately predicts protein structures and has paved the way for other peptide design architectures such as AlphaDesign [24]. DeeProtein and ProDCoNN are also two examples of deep learning-based methods that have achieved state-of-the-art performance in predicting protein function and structure [43,44]. Other notable methods such as RoseTTAFold, trRosetta, DeepCNF, DuetDis, and PSICOV have also been developed to predict the 3D structure of peptides with high accuracy [6,45,46,47,48]. These methods have the potential to greatly enhance our understanding of biological systems and enable the design of peptides with specific structures and functions.

#### 2.3.1. AlphaFold and AlphaDesign

AlphaFold is a deep learning-based method for predicting protein structures developed by researchers DeepMind Technologies [49]. It uses a deep neural network trained on a large database of known protein structures to predict the 3D structure of a protein from its amino acid sequence, incorporating multiple sequence alignment information and modeling the distances between pairs of amino acid residues [49]. The predicted structures are highly accurate and have important applications in understanding protein function, drug design, and disease treatment. The most recent version of AlphaFold, namely AlphaFold2 was benchmarked for its accuracy in predicting peptide structures and was found to perform comparably or better than other methods that were designed for the same purpose [50]. To this point, a recent study performed by Tsaban and colleagues demonstrated the ability of AlphaFold2 to accurately predict protein-peptide interactions without prior training [51]. Specifically, AlphaFold2 was demonstrated to accurately predict high-affinity binding interfaces between peptides and proteins, with or without available binding pockets. Remarkably, AlphaFold2 can also predict peptide binding sites upon protein conformation change. Furthermore, the length of assessed peptides was found to not interfere with docking performance [51].

A recent study used AlphaFold2 to generate a peptide that targets a biomarker of diabetes [52]. In this study, the authors used AlphaFold2 to link a skeletal muscle-targeted peptide with a mutated FGF1 protein [FGF1^ΔHBS^]. This variant retains metabolic activity while preventing mitogenic interactions with FGFR. Using this peptide-linked FGF1 variant, the authors found that the AlphaFold2-generated peptide was critical to associate FGF1 to skeletal muscle [52]. This notion is supported by several *in-vivo* and *in-vitro* assays including fluorescence microscopy, protein-based assays, and *in-vivo* mouse monitoring assays. Before this study, long-term application of the naïve FGF1^ΔHBS^ to mice resulted in several major disadvantages including weight loss, loss of appetite, and death. With the new AlphaFold2 linked peptide—FGF1^ΔHBS^ conjugate, these side effects were reported to be reduced [52].

Another recent example of using AlphaFold2 to develop peptide therapeutics is the development of the N1S peptide [53]. In this study, Modi and colleagues used AlphaFold to develop peptides that would interfere with the Nrf2-MAFG interaction. Specifically, three peptides of 16 amino acids in length were designed named N1S, N2S, and N3S. Upon activity screening, only the N1S peptide displayed activity while showing no hemolytic properties. In terms of peptide affinity, the authors found that the N1S peptide has a dissociation constant (Kd) of 337 nM and can reliably prevent Nrf2-MafG heterodimerization [53]. Thus, using AlphaFold to develop peptides or peptide linkers appears to be an effective and safe option to generate peptide therapeutics.

AlphaDesign is a similar method that predicts the 3D structure of a peptide from its amino acid sequence, using a generative adversarial network architecture that is trained on a dataset of protein structures and incorporates AlphaFold within an optimized design process [23]. AlphaFold2 can predict peptide structures with high accuracy but has limitations in predicting certain structural features [54]. Therefore, while AI architectures can be powerful tools for predicting peptide structures, additional steps may be necessary to analyze and validate the results.

#### 2.3.2. Additional Neural Network-Harnessing Technologies

In recent years, several deep learning-based methods have been developed for predicting the functions and structures of proteins. DeeProtein is one such method that utilizes residual and fully convolutional neural network architecture for the multi-label classification of protein sequences into 539 functional classes [39]. This network was trained on a large dataset of protein sequences and can be used for functional dissection and engineering of proteins. DeeProtein can predict the 3D structure of peptides by inferring its 3D structure based on the predicted functional class of the protein sequence [43]. ProDCoNN is another method for designing protein sequences that fold into a given 3D structure [44]. This method is based on a 9-layer 3D deep convolutional neural network that takes atomic coordinates and types around a residue as input. The convolutional neural network layers in ProDCoNN are specifically designed to capture structural information at different scales, such as bond lengths, angles, torsion angles, and secondary structures [44]. The method achieved state-of-the-art performance when tested on large numbers of test proteins and benchmark datasets, after being trained on a very large number of protein structures.

DeepCNF is an extension of Conditional Neural Fields (CNF) that integrates conditional random fields and shallow neural networks, allowing it to model complex sequence-structure relationships and interdependencies between adjacent secondary structure labels [47]. Experimental results demonstrated that DeepCNF outperformed popular predictors and could also predict other protein structure properties [47]. DuetDis is a similar CNF that utilizes duet feature sets and deep residual networks with squeeze-and-excitation to predict fine-grained distances between residues with long sequence separations [48]. It combines features from whole genome/metagenomic databases to minimize information loss and improve prediction performance. DuetDis outperformed peer methods in terms of accuracy, reliability, and robustness [48]. Another tool, Protein Sparse Inverse COVariance (PSICOV) predicts residue-residue contacts by accurately discriminating between direct and indirectly coupled mutation correlations in multiple sequence alignments, which may have significant impacts on the prediction of structure and function [6].

#### 2.3.3. Rosetta

RoseTTAFold is a highly accurate method for predicting protein structures that uses three tracks representing the amino acid sequence, inter-residue distances, and 3D coordinates of the protein [45]. By training on smaller peptide fragments and averaging predictions, it surpasses currently available techniques in accuracy, enabling it to assist in solving challenging MR problems and improve borderline cases. Recent work also demonstrated that tr-Rosetta can be used as a computational method for predicting the 3D structure of a protein from its amino acid sequence [46]. It is an extension of the Rosetta protein modeling software and incorporates deep learning techniques to refine the predicted structures. This method has shown high accuracy in predicting protein structures and has outperformed other state-of-the-art methods in terms of accuracy [46]. It can be used to predict the 3D structure of peptides from their amino acid sequences, which can be useful in designing peptides with specific structures and functions.

Rosetta was previously used to generate small peptide inhibitors against the SARS-CoV-2 (the virus responsible for the COVID-19 pandemic) spike protein and to BRAF—a member of the RAF-kinase family [55,56]. At the outset of the COVID-19 pandemic, computation was harnessed to produce a multitude of potential therapeutics to interrupt the SARS-CoV-2 lifecycle [55]. In this study, Rosetta was used to produce peptides *ab-initio* against the SARS-CoV-2 spike protein. Specifically, the authors report that thousands of potential peptides were predicted using Rosetta. From an initial pool of 2.5 thousand peptides, a combination of microarray screening and ELISAs reduced this pool to 4 peptide candidates [55]. These candidates were then experimentally validated using a bio-layer interferometry assay, in which the dissociation constant for each peptide was determined to be between 100 and 250 nM [55]. This study, therefore, demonstrates the ability of Rosetta to design high-affinity peptides to a consequential human pathogen biomarker.

In another study, Rosetta was used to design several peptides against the sequence that is thought to mediate the BRAF PPI [56]. Using the highest-ranked peptides, the authors performed experimental validation through ELISAs and Co-immunoprecipitations between BRAF and the peptides. They found one peptide—subsequently named braftide—displayed an IC_50_ of 364 nM to WT BRAF and 172 nM against oncogenic BRAF [56]. Additional experimental validations between braftide and BRAF demonstrate the ability of Rosetta to accurately predict a high-affinity peptide to a human cancer biomarker through *in-vitro* experimental validation [56].

The computational methods discussed thus far can predict the complex and diverse structures of peptides, such as α-helices, β-sheets, and disulfide-rich peptides with high accuracy. The development of these methods has been driven by the need to understand the structure and function of peptides, which play important roles in biological systems [23]. These methods have the potential to revolutionize the field of structural biology, by enabling researchers to predict the structures of complex molecules quickly and accurately, and to better understand their functions in biological systems. While challenges remain, the ongoing development of these methods is likely to lead to further breakthroughs.

## 3. Designing Peptide Mimics

Protein mimicry refers to the process of manipulating and designing peptides to mimic the structural and functional characteristics of small molecules. Peptides designed in this manner can therefore play crucial roles in various physiological processes, and their manipulation has the potential to serve as a key strategy for the development of novel drugs and therapies for various diseases [14,57]. Computational tools can be utilized to further refine the design of peptide structures, enabling the development of mimics with enhanced activity and specificity [14]. This approach offers a promising strategy for gaining a deeper understanding of the complex pathways involved in disease and identifying specific target areas for therapeutic modification. Protein mimicry can be achieved through different approaches, such as DeNovo and structure-guided design techniques. These techniques have been applied to a wide range of applications including the stabilization of protein structures, development of novel drugs, design of mimetics for vaccine production, and the production of antibiotics [57,58]. Mimicry, therefore, is a powerful tool for understanding and manipulating complex biological processes, with the potential to revolutionize the development of new therapies for a wide range of diseases.

### 3.1. DeNovo

The DeNovo mimic approach is a promising strategy for designing novel molecules with potential biomedical applications. One such application is the development of bone morphogenetic protein (BMP) mimics, which are peptides that can induce bone and cartilage formation [58]. Researchers have successfully used the Rosetta software suite to design two DeNovo protein mimics of BMP-2, which were highly stable and effective at inducing bone growth *in-vivo* [58]. This approach holds potential for the development of therapeutics for bone disorders. In addition, the DeNovo mimic approach has been used to design Spliceostatin E (SPE)—a peptide that selectively inhibits the activity of Fibroblast Growth Factor Receptor (FGFR) splice variants that are overactive in cancer cells [59]. The designed SPE protein targets the FGFR heparin binding site and selectively inhibits the activity, having little to no effect on other isoforms. This approach holds promise for developing targeted therapies for cancer treatment [59].

The DeNovo mimic approach has also been used to design a modified CRISPR-Cas9 system, dCas9, which targets the Polycomb Repressive Complex 2 (PRC2) inhibitor to specific DNA sequences to facilitate gene expression [60]. dCas9 was shown to decrease Myc expression and increase H3K27me3, a marker of PRC2-mediated gene repression, demonstrating its potential for use in gene regulation and gene therapy. These studies demonstrate the potential of the DeNovo mimic approach for designing peptide therapeutics for biomedical applications, including the development of new therapeutics for diseases such as cancer and bone disorders.

### 3.2. Structural-Guided Design

Structural-guided design is an approach that combines topological data analysis and machine learning to design stable and functional protein variants [17]. One notable example is the Persistent Spectral Theory-guided Protein Engineering method, which has outperformed other state-of-the-art methods and successfully identified a stable cytochrome c variant while improving the stability, flexibility, and enzymatic activity of various protein systems [17]. An excellent review by Gupta and colleagues on this technique describes several examples of peptides generated using this approach [14].

Recent use of this technology has yielded several peptides that can interfere with consequential drug targets. A notable example is recent work performed by Kaur and colleagues, who used this technology to develop peptides that target the RNA polymerase of the human pathogen *Mycobacterium tuberculosis* [61]. Specifically, Kaur and colleagues developed several peptides that were designed to specifically target conserved residues within the RNA polymerase-transcription factor complex [61]. Of the several peptides assessed, they found that two seem to display activity, as measured by transcription efficiency.

During the COVID-19 pandemic, there was a necessity to rapidly develop a wide range of therapeutics to prevent infection. Early in the pandemic, the ACE2 receptor became not only the target, but also the template for the design of many of these therapeutics, as ACE2 is responsible for the entry of the SARS-CoV-2 virion into the host cells. To this point, a recent study performed by Karoyan and colleagues used protein mimicry to develop peptides that block SARS-CoV-2 entry into pulmonary cells [62]. In this study, the authors identified 20 residues within the N-terminus of the ACE2 receptor that came into close contact (4A) with the SARS-CoV-2 spike protein [62]. This information was then used to develop several peptides that were specifically designed to avoid immunogenicity while being optimized for high-affinity binding to the viral spike protein [62]. Upon testing the effectivity of these peptides, the authors found that two peptides appeared to display the ability to block viral infection, namely P7 and P8. These peptides informed the development of additional peptides (P9 and P10). Peptides P8–P10 were then shown to display a high affinity of to the SARS-CoV-2 spike protein in a dose-dependent manner. In terms of *in-vivo* applications, the authors found that these peptides displayed a 100% efficiency in reducing viral infection at a concentration of 1 µM, highlighting the affinity of these designed peptides [62].

In addition to human pathogens, structural-guided peptide design can also be applied to mammalian proteins. One such example is a recent study performed by Cardote and Ciulli, which used this technology to design peptides against the interaction between E3-ligases and adaptor proteins [63]. The purpose of this study was to determine whether small peptides can be used to target Elongin C, a component of an adaptor subunit. They found that two peptides generated with this technology were able to bind to the contact site between Cullin 2 and Elongin C, termed with the EloC site. Specifically, these two peptides displayed an efficient dissociation constant, suggesting that binding did take place [63].

Another recent study assessed whether structural-guided peptide design can improve the T-cell receptor (TCR) specificity [64]. TCRs recognize antigens presented by antigen-presenting cells, in which the antigens are usually small peptides [64]. To this point, TCRs display low specificity culminating in high cross-reactivity. Thus, the purpose of this study was to use a structural-guided design to enhance the specificity and reduce the cross-reactivity of TCRs. The authors found that TCRs generated using structural-guided design displayed a higher specificity to target peptides, with the strongest affinity being a 400-fold increase in affinity to a MART-1 peptide relative to WT TCR [64]. Thus, these studies provide further evidence for the efficiency of peptides generated through structure-guided design.

Another area of research involves the binding interactions of small molecules with RNA G-quadruplexes, which play important roles in gene regulation and cellular processes. By identifying structural features of small molecules that selectively stabilize RNA G-quadruplexes, researchers have the potential to develop new drugs that target these specific structures [65]. Lastly, algorithms such as TopoBuilder and RITA are used in DeNovo protein design. TopoBuilder generates diverse protein folds and pockets through fragment assembly and scoring functions, with several proteins capable of binding to small molecules *in-vitro* studies [66]. RITA, a computational tool used for training generative protein sequence models using large datasets and advanced computational resources, has potential implications for the development of new and more efficient methods for peptide design and drug discovery [67].

## 4. SM-Focused Design

The modularity of SMs offers a broad range of potential drug targets, thereby allowing for the generation of multifunctional peptides with different specificities. Due to being highly conserved across species, SMs are useful for the development of peptide therapeutics with reduced efficacy gaps in animal models [19,20]. Additionally, SMs can be used as a scaffold to determine protein interactions quickly and efficiently, making them valuable for screening potential drug candidates [68]. To this point, we previously developed the Protein–Protein Interaction Prediction Engine (PIPE) which can accurately predict protein interaction motifs between several proteins [68,69,70,71]. This is achieved by relying on an input dataset from protein databases such as BioGrid, which itself contains 40 thousand interacting protein pairs. The underlying principle behind PIPE and how this algorithm has evolved into generating inhibitory peptides is discussed below.

### 4.1. Motif-Based Protein–Protein Interaction Prediction

PIPE leverages SMs found in query proteins that mediate other known interactions to generate its predictions [68]. Specifically, PIPE scans through the amino acid sequence of query protein A and compares this to previous interaction data within protein databases to find similar motifs (Figure 2). Once a matching sequence/subsequence/motif is found (between the query sequence and a known protein), all known interactors of the protein containing the homologous motif are noted. This process continues until the interaction network has been completely scanned for matching subsequences with query A. Next, query protein B is compared to the protein network associated with matching motifs from query A. Every time a subsequence from query protein B is found to interact with a protein partly homologous to A, the probability of an interaction between A and B mediated by these subsequences increases. These scores are kept in a 3D result matrix, where a plot of protein A against protein B’s sequences is extruded in the third dimension whenever an interacting pair of subsequences is found. The PIPE data matrix is then used to calculate the fitness (or specificity) of each amino acid sequence by comparing their affinity to targets vs. non-targets [68]. The PIPE algorithm was later expanded to accommodate global PPI analysis. In 2008, it was used to elucidate the first computationally predicted genome-wide PPI analysis in a cell [69]. It was later used to predict inter-species PPIs [71].

De Novo is a comparable PPI prediction algorithm that leverages SMs in primary sequences to evaluate host-virus interactions [9]. Unlike PIPE, this algorithm is suited towards applications where there are few known interactions to train it. Instead, De Novo predicts interactions using the physiochemical properties of the host proteins and their interactions with other known viruses [9].

Another tool that can be used to predict protein-peptide interactions is GalaxyPepDock [72]. This tool incorporates a database of known protein-peptide interactions to inform novel peptide docking predictions. In terms of accuracy, GalaxyPepDock outperformed similar tools such as PEP-SiteFinder and PepSite in identifying potential peptide binding sites on target proteins [72].

PIPE was expanded upon in 2011 with the development of PIPE-Sites, which aimed to analyze the topology of the result matrix output by PIPE to automate PPI analysis, enabling the exploration of SM discovery at the scale of entire proteomes [70]. PIPE-Sites first determines the peak(s) in the result matrix, which represent highly occurring amino acid pairs in interacting protein pairs, then measure their cross-sectional area to find the size of the binding site. PIPE-Sites successfully predicted previously documented PPI binding sites in yeast from a PIPE result matrix with very little discrepancy from the known regions. PIPE-Sites discovered nearly 1000 non-annotated potential interaction sites among a dataset of 14,438 interacting yeast proteins, which was obtained from a previous screen [69]. PIPE-Sites expands upon PIPE by automatically analyzing PIPE data to provide potential SM between interacting pairs, which may be used to conduct exhaustive whole-proteome interaction searches.

Currently, several additional databases and tools are specific to SM annotation and discovery. DIscovery of LInear MOTifs (DILIMOT) is an SM discovery tool that utilizes protein FASTA sequences as input in addition to using BLAST to find orthologous groups and MUSCLE for multiple sequence alignments within intrinsically disordered regions. These are then analyzed by a pattern-matching algorithm to find motifs [73]. The Short Linear Motif (SLiM)Search and SLiMFinder are similar tools that use BLAST for sequence alignments to find short recurring sequences that may mediate PPIs [74], while others such as the Protein–Protein Interactions Domain Miner infer SMs or domain-domain interactions, from many sources of PPI data (KBDOCK and 3did [75,76]).

In 2014, PIPE was expanded to increase its efficacy by parallelizing it, resulting in Massively Parallel-PIPE (MP-PIPE) [6]. MP-PIPE aimed to enable whole-proteome analysis in organisms with complex interactomes, providing more applicable information to biomedical research, particularly in peptide therapeutics [77]. By scanning the human proteome, MP-PIPE predicted 172,000 PPIs, of which 133,000 were newly discovered, quadrupling the known human interactome knowledgebase [77]. MP-PIPE only accesses protein sequence data and incorporating more information such as structure and cellular localization might improve its accuracy, such as matching gene-ontology terms to increase confidence in predicted PPIs. We used MP-PIPE was used to detect novel interactions in the breast cancer pathway, discovering three novel interactors, CDK3, AURKB, and SMC1B, and observed two candidate mutations participating in different interactions which might lead to therapeutic resistance [8]. A sensitivity of 23% might be further enhanced by an increase in data quality and further optimization/parallelization. Incorporating accurate structural data is another method to increase the efficiency of this algorithm to predict interactions at a global level.

The latest version of PIPE, namely PIPE4, utilizes interaction graphs to solve the interactomes of poorly annotated organisms [78]. PIPE’s use of interaction graphs introduces a bias towards PPIs like those in the host organism. While this bias has not been problematic for discovering PPI data for well-studied organisms [8,69,79,80,81], it can pose a challenge for investigating PPIs for an emerging pathogen for which little interaction data is available. PIPE4 addresses this issue by annotating inter-species interactomes to elucidate disease pathogenesis and explore interactome evolution. To this point, *Heterodera glycines* is an agricultural pathogen that affects soybean, the latter having three billion putative PPIs [78]. We used PIPE4 to predict interactions between soybean and *H. glycines* using cross-species proxies Arabidopsis thaliana and Caenorhabditis elegans [78]. This approach allowed us to accurately annotate the host-pathogen PPIs between soybean and *H. glycines*. A post-experimental validation called Reciprocal Perspective for Improved Protein–Protein Interaction Prediction was conducted to decrease the false-positive rate in PIPE4 [82]. This validation considers a reciprocal relationship in protein pairs instead of treating one as a binder and another as a ligand, leading to fewer false positives. We recently leveraged PIPE4 in the prediction of human-Soybean PPIs, with the goal of predicting PPIs involved in human-soybean allergies [83]. We found several novel PPIs that might be consequential to human health and allergy between the two organisms [83].

### 4.2. SM-Based Peptide Design

The *In-Silico* Protein Synthesizer (InSiPS) is a novel method for designing inhibitory peptides capable of selectively binding and inhibiting specific target proteins in living organisms [10]. The process involves generating new peptide sequences by modifying a pool of randomly generated amino acid sequences using copy, mutate, and crossover operations, and then evaluating the fitness of each peptide using PIPE which ranks peptides based on their predicted affinity to the target protein (Figure 3). The peptides with the highest fitness scores are used to create a new pool, and this process is repeated until the top-scoring peptides’ fitness plateaus. As the binding fitness is derived from PIPE, the generated peptides interact with the target protein based on the interaction between linear motifs as described above.

An important attribute of InSiPS is the negative selection of the peptides [10]. For each cycle, there is a negative selection against a predefined set of non-target proteins. Each peptide is independently evaluated for its binding ability to the non-targets. Therefore, only those peptides that bind to the target proteins (but do not interact with the set of non-target proteins) are selected and used in the next generation. After each round of selection, the pool of peptides is further enriched with peptides that interact with the target protein while simultaneously avoiding non-targets [10]. The set of non-target proteins are different for each experiment. For example, if the target protein is a human cell surface receptor, then the set of non-targets may constitute other human cell surface proteins. Performing this negative selection enhances the specificity of the designed peptides to the target protein and is, therefore, an important and unique attribute of InSiPS [10]. As peptides generated by InSiPS are designed based on naturally occurring interaction motifs, the chances of them interfering with PPIs and consequently having a biological function (for example, deactivation) is very high.

To this point, a proof-of-concept experiment demonstrated that InSiPS can design peptides that inhibit the function of yeast proteins [10]. The peptides generated by InSiPS were tested against three target proteins *in-vivo*: Rmd1 (cytosolic sporulation protein), Pin4 (DNA damage repair protein), and Psk1 (serine/threonine protein kinase). The anti-Pin4 and anti-Rmd1 peptides displayed inhibitory activity similar to the phenotypes associated with PIN4 and RMD1 deletions, respectively. Specifically, we found that WT yeasts treated with the anti-Pin4 peptide displayed a sensitivity phenotype to 0.1 M of arsenite in a manner similar to WT untreated yeasts. These results were further supported by a yeast-2-hybrid assay which demonstrated the ability of both anti-Pin4 and anti-Psk4 peptides to bind to Pin4 and Psk1. We also directly evaluated the binding affinity for these peptides using an *in-vitro* walking array. We found that anti-Psk1 displayed a Kd of 2.2 nM, indicating a strong interaction between the peptide and the target protein. In these experiments, anti-Rmd1 failed to demonstrate increased sensitivity to β-mercaptoethanol or L-1,4-dithiothreitol, as would be seen in an *rmd1Δ* strain.

Recently, InSiPS was utilized to design peptides against SARS-CoV-2 surface protein S [84]. To accomplish this, two different regions of the S protein were selected as independent protein targets: the receptor binding domain and the S1/S2 region, both of which are implicated in virus entry into the host cell [85]. Human cell surface proteins were selected as non-target proteins. Peptides designed in this way were evaluated for their abilities to bind and hence detect the S protein. We further demonstrated that the designed peptides can be used for both the capture and detection of the S protein from a protein mixture using an ELISA analysis. Of the 10 designed peptides, 6 were found to have detectable levels of binding to the target proteins. As little as 0.1 ng of the S protein in 1 mL of protein mixture was detected by these peptides. The applicability of one of these peptides for real-time COVID-19 diagnostics was evaluated using surface plasma resonance analysis. The Kd value of that peptide was measured to be more than 10-fold lower than that of the natural S protein interacting partner, the human ACE2 receptor protein.

In a manuscript under preparation, additional peptides designed in this way were evaluated for their therapeutic abilities to interfere with the SARS-CoV-2 life cycle. Of the 10 peptides evaluated, four appear to interfere with the binding of the viral S protein to its human host ACE2 cell receptor in an *in-vitro* assay as well as in an in-vivo biosensor experiment (unpublished results). One of these peptides reduced the replication of the SARS-CoV-2 virus by more than 75% in cell culture analysis.

Although InSiPS uses primary sequences as inputs (which allows for the rapid design of peptides in addition to enabling non-computationally intensive evolutionary approaches), the length of the peptide is an important factor to consider during design [10]. Larger proteins can have increased occurrences of complementary amino acid sequences, increasing their affinity and specificity to the target protein. This makes further analysis and experimentation difficult and expensive. Furthermore, a larger protein has a tertiary structure that is inherently harder to predict and could potentially fold in a manner that sterically reduces its affinity to the target [10]. In addition, the interactions predicted by InSiPS have a certain degree of error that must be considered prior to experimentation, as the peptides may not functionally bind and inhibit their target. However, since the peptides are designed based on interaction sites it is likely that the binding peptides will be functionally relevant. As such, while InSiPS is limited to target protein size and potential binding errors, it also exhibits high potential for developing SBPs with high affinity and specificity to target proteins.

## 5. Current Advantages and Future Challenges of *In-Silico* Peptide Design

Synthetic peptides have a significant advantage over small-molecule drugs in that they are structurally complex which can enable a single peptide drug to interact with many of the target binding sites required for activation or deactivation. One example of a highly successful peptide drug is insulin [86]. Various techniques are available to design therapeutic peptides including primary and tertiary peptide structure-based design. The latter approach involves the use of 3D protein structures and computational docking to identify potential peptide candidates that can bind to specific targets. However, this approach has limitations due to the availability of high-resolution protein structures and the accuracy of docking algorithms. Though there are numerous *in-silico* approaches to designing and generating peptide therapeutics each with benefits and drawbacks, two relevant questions emerge pertaining to the *in-vivo* application of synthetic peptides. Do synthetic peptides generated designed *in-silico* display bioactivity and can synthetic peptide toxicity be predicted?

### 5.1. Peptide Bioactivity and Toxicity

Several *in-silico* tools have been developed that can predict the biological activity of synthetic peptides [87,88,89,90,91]. One such too is the predictor of the antihypertensive activity of peptides (PAAP) which, as its name suggests, was developed to predict whether peptides displayed antihypertensive properties [87]. PAAP builds on a previous model which predicted the inhibitory activity of small peptides [88] and utilizes a random forest algorithm. To confirm the ability of PAAP to predict peptide activity, the model was passed through a 10-fold cross-validation, leave-one-out cross-validation, and was compared to a previously established prediction model AHTPIN [89]. These validations confirmed the ability of PAAP to predict the antihypertensive activity of peptides.

Another peptide prediction tool is HemoPred. With the rapid and widespread development of synthetic peptide therapeutics, understanding whether synthetic peptides can induce hemolysis is essential. HemoPred was developed for this purpose [90]. This algorithm considers three sequence features of synthetic peptides including the physiochemical properties, the peptide composition, and the dipeptide composition [90]. Various types of cross-validation confirm the ability of HemoPred to outperform previous approaches, such as HemoPI, SVM, and DT [90]. To this point, HemoPred was found to outperform HemoPI, in terms of mean accuracy of the prediction, by 2–3% points. This was observed across multiple validation tests, suggesting that HemoPred is a highly accurate *in-silico* model for predicting the hemolytic activity of synthetic peptides.

In terms of deep learning algorithms, a recent tool that uses deep learning to predict the anticancer activity of peptides (DeepACP) was developed [91]. Specifically, DeepACP uses three deep learning techniques including CNN, recurrent neural networks (RNN), and CNN-RNN. Of the three, the RNN-based approach held the most promise for the accurate prediction of anticancer properties, as this approach demonstrates a precision score of 89.5%, an 83.9% F value, and an 84.9% accuracy score [91]. Compared to a CNN-based approach, the RNN scores demonstrated a 4.8%, 1.6%., and 2.2% increase in performance respectively. Though these results were not validated *in-vitro*, the high accuracy of this *in-silico* model holds promise in the prediction of the anticancer activity of synthetic peptides [91]. These tools were developed due to synthetic peptides displaying various levels of immunogenicity—a major barrier to the large-scale adoption and application of peptides as therapeutics [92,93]. In addition to their hemolytic activity, peptides can also display several specificity-related side effects such as the inappropriate association with immune receptors [94,95,96]. To this point, the field of immunoinformatics has produced a few tools to predict the immunogenicity and toxicity of synthetic peptides [94,95]. One such example is SYFPEITHI—a prediction model that can screen small peptide libraries against immune receptors [96]. This *in-silico*-based screening approach relies on the rank-ordered presence of anchoring amino acids as well as amino acids that are likely to mediate an interaction [96]. Though not the focus of this review, tools have also been developed to predict the interaction between larger peptides and immune receptors, such as NetMHC and SMM [97,98,99,100].

Though tools are available to predict peptide toxicity through inappropriate interactions, peptides can still bind to other targets if not developed with an anti-selection [10,94,95,96]. To this point, sequence-based peptide design relies on the identification of SM in protein sequences that mediate PPIs [10]. Additional sequence-based algorithms may use machine learning to infer physiochemical properties from these sequences [101]. Regardless of the protein-target interaction scoring methodology used, the designed protein often undergoes affinity maturation. While there are many approaches to developing therapeutic peptides, InSiPS is one of the few genetic algorithms capable of designing a peptide with high specificity to a given protein target [102]. The negative selection against a predefined set of non-target proteins is a unique attribute of InSiPS. Such a negative selection significantly increases the specificity of the designed peptides and may contribute to reduced toxicity and the side effects of the peptides that are designed for therapeutic purposes.

### 5.2. Barriers to FDA Approval of Computationally Designed Peptides

Generating computationally designed peptides for *in-vivo* applications is in its infancy [50,103]. Therapeutic peptides have several advantages over small-molecule drugs, including high selectivity and binding affinity without the downside of decreased stability and increased immunogenicity [104]. Despite their potential, peptides only account for approximately 5% of the global pharmaceutical market in 2019 [2]. Though we have discussed several examples of *in-vivo* and *in-vitro* applications of designed peptides, many barriers still exist that prevent their widespread adoption, the most critical being approval by the Food and Drug Administration (FDA). To our knowledge, there is no FDA-approved peptide drug that was originally designed through computational approaches. It is noteworthy to briefly discuss the current barriers of such synthetic peptides that impact their FDA approval and thus large-scale adoption as therapeutics.

Predicting and modeling peptide structures is a complicated task, as peptides can fold into complex three-dimensional structures [76]. This can impact the ability to accurately predict their structure based on computational methods alone. Often, experimental validation is required to confirm the predicted structures [10,51,52,53]. To this point, nearly all of the examples discussed generate several candidate peptides, with few only displaying an effect. This raises a second potential barrier to widespread adoption. Designing peptides that can selectively bind to the desired target, while avoiding off-target interactions is difficult [10]. Given that peptide therapeutics can range in length (here we have discussed peptides that are as small as 16 amino acids, and as large as 40), it is likely that these small amino acid chains can interact with a multitude of non-targets. With the exception of InSiPS, we are not aware of other algorithms that have a built-in negative selection to prevent these non-target associations.

Another barrier is the biological stability of synthetic peptides. Like every biomaterial, peptides are subject to enzymatic degradation [105]. Whatsmore, the fact that synthetic peptides are foreign to the host also limits their stability and bioavailability. While computational tools can aid in predicting peptide stability, only experimental validation can confirm whether synthetic peptides are stable and bioavailable. Peptide half-life refers to the rate at which these peptides remain stable and bioavailable [106]. To this point, renal clearance of peptides can occur within minutes [106]. Certain technologies are developed to enhance peptide stability [107]. Computational tools also exist to further enhance these aspects of synthetic peptides, though again the only real way to confirm peptide half-life is through experimental validation which is a laboratory-intensive process. In terms of delivering bioactive and nontoxic peptides into the host, peptides also have low membrane permeability and oral bioavailability [108]. As a result, 90% of current peptide therapies require injection which poses a challenge for medical adherence [108].

A final barrier to large-scale adoption of computationally designed peptides is manufacturing and cost-effectiveness [92,109,110,111,112]. In terms of manufacturing, once a computationally designed peptide displays a desired effect, be its inhibition of a PPI or detection of a protein biomarker, scaling up production during manufacturing can pose challenges [92,109,110,111,112]. For instance, ensuring peptide purity, consistency, and quality control at a large-scale may require additional optimization. Though this is an important point to consider, it is thought of as a minor point as in certain cases these issues can be readily resolved. In terms of cost-effectiveness, developing and manufacturing peptides can be an expensive venture, especially ona small scale. In basic research, synthesizing peptides can cost thousands of dollars with only one or two peptides displaying some sort of effect [110,112]. This is a major barrier to studying, designing, and generating peptides. Though when shifted to a large-scale manufacturing environment, the cost-effectiveness of producing peptide therapeutics can be rapidly consolidated. Thus, the barrier to peptide therapeutic development in terms of cost is front-loaded in basic research and academia thereby in certain cases preventing widespread investigations into studying and developing peptide therapeutics.

### 5.3. Applications of Therapeutic Peptides and the Future Peptide Design

We have discussed several methods and tools to develop and predict the activity of synthetic peptides. How then can these peptides be applied? Several examples have been discussed, such as anticancer and antiviral peptides [62,84,91]. Aside from the biological and cellular applications of peptide therapeutics, these synthetic peptides also hold promise for developing new diagnostic tools. For example, von Willebrand disease (VWD) and platelet-type VWD have nearly indistinguishable symptoms. This is due to both diseases affecting one of two proteins in a critical PPI between platelet glycoprotein 1b α and von Willebrand factor proteins [113]. Specifically, a mutation in the β-sheet of GP1ba results in an excessive interaction with VWF [113]. Generating synthetic peptides that are specific to one of these proteins can therefore eventually result in a diagnostic test to determine which biomarker is being affected. Another diagnostic tool can be developed relating to microbial pathogens. We previously used InSiPS to develop a proof-of-concept ELISA assay to detect the SARS-CoV-2 spike protein S [84]. Thus, there is the possibility that synthetic peptides can be developed against key biomarkers of microbial pathogens, eventually culminating in several rapid tests for bacterial and viral disease using the same ELISA-like approach. Similarly, ELISA-based assays to measure protein contents of any protein of interest can be made using designed peptides that specifically bind to that protein.

Although the initial version of InSiPS was computationally intensive, the recently modified version of the algorithm has a 100-fold improved speed. Using the latest version of InSiPS, we plan to initiate a very ambitious project—developing two peptides for every essential human protein. Our goal is to expand this approach to the entire proteome of humans and eventually other organisms (mice, yeast, etc.). With every peptide successfully designed by InSiPS, the algorithm undergoes further training, thereby becoming faster. This is an important feature, as the demonstrates the power and process of AI-designed therapeutics as becoming increasingly efficient. Thus, it is likely that AI will become an integral technology in the development of disease therapeutics. Instead of solely relying on traditional therapeutic development, the use of AI in therapeutic design has the potential to dramatically increase the speed and efficiency of peptide therapeutics.

## Figures and Tables

**Figure 1 genes-14-01194-f001:**
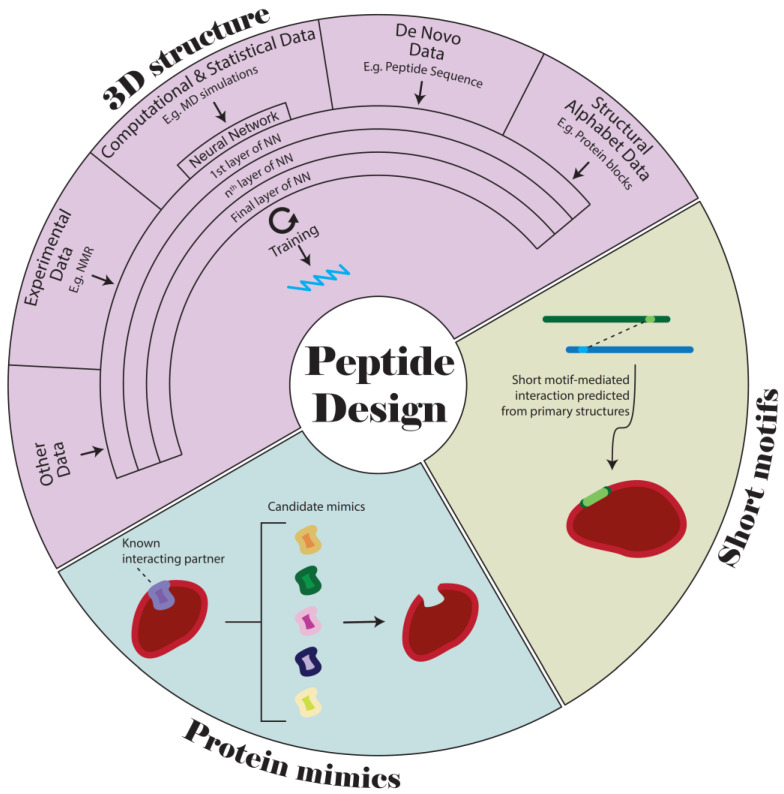
The three primary technologies used to design small peptide therapeutics. The 3D structure comprises a large portion of the technologies used to generate peptide therapeutics. This structural technology utilizes five sources of data: experimental, statistical, de novo, alphabet, and other sources. These inform neural networks to train the structural-based technologies to develop peptide therapeutics. Protein mimicry involves generating peptide candidates derived from known interaction partners. Candidate mimics are then tested against interaction partners. Short motifs (SMs) comprise the final technology used to generate peptide therapeutics. This involves developing peptides from previously known interaction motifs, which are then used to predict and generate small peptide therapeutics. All three technologies are reviewed in this manuscript.

**Figure 2 genes-14-01194-f002:**
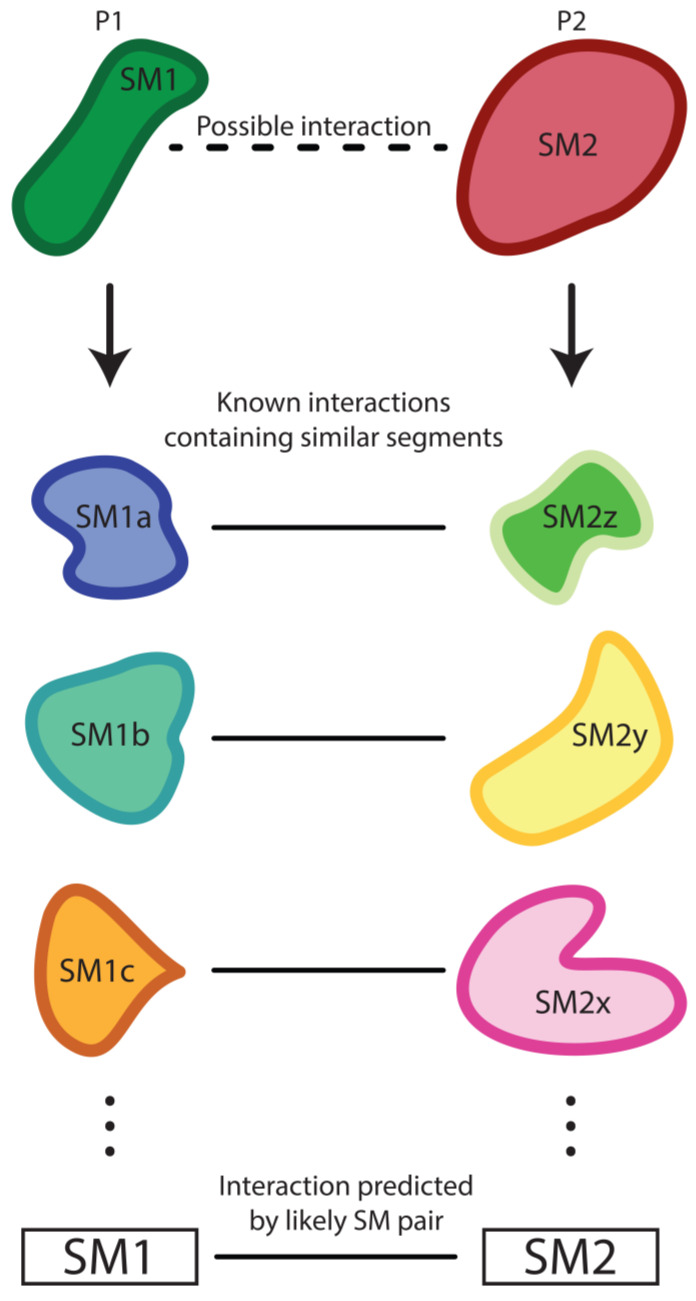
The theory underlying the PIPE algorithm. PIPE predicts PPIs through leveraging known interactors that contain similar segments. SMs that mediate known interactions are used to train the PIPE algorithm to predict new interactions. The prediction output is ranked based on the probability of the two SMs interacting. The probability of an interaction is informed by the known interaction data. PIPE has demonstrated the ability to predict novel PPIs in yeast and humans.

**Figure 3 genes-14-01194-f003:**
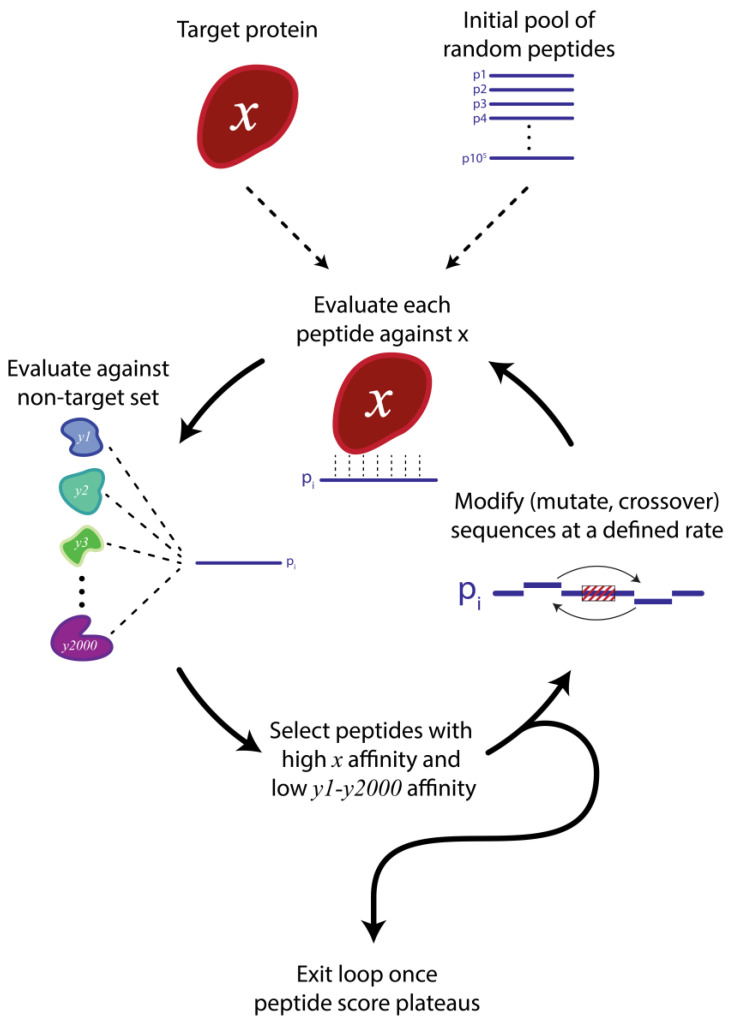
InSiPS generates peptides specific to a wide range of protein targets. First, an initial pool of mixed random peptides *in-silico* is generated. These peptides are then evaluated against the target protein to determine whether they interact. At the same time, the peptides are evaluated against a pre-defined set of non-target proteins, to ensure high target specificity. The highest-ranked peptides that bind the target and avoid non-targets are then modified through mutations and crossover applications to enhance target affinity. This process is repeated until the peptide score (i.e., the highest predicted target affinity) plateaus. At this point, the peptides are ready for further *in-vitro* and *in-vivo* analysis.

## Data Availability

Not applicable.

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
