# Peer review of "Peptides of a Feather: How Computation Is Taking Peptide Therapeutics under Its Wing"

_genes, 2023, doi:10.3390/genes14061194_

Round 1
Reviewer 1 Report
Data-driven computational methods are in silico tools that promise to revolutionize the discovery and development of new therapeutic candidates. Due to the high therapeutic value of peptides, these tools are considered strategic allies for significant advances in peptide-based therapy and implications for modern medicine. The present review, is an excellent piece of work that combines a series of in silico strategies focused on the design and development of therapeutic peptides. This review summarises important advances in the field and is well-written. Here, I detail some suggestions.
1. The study is focused on peptide therapeutics. However sometimes the authors mentioned proteins or protein therapeutics. There are relevant differences between them. I suggest the authors does not change the focus of the study. My main recommendation is only describe peptide therapeutics. Please see lines 34, 46, 58 and so on.
2. Some sentences need to be supported by references. For instance, lines 45-50, 50-52, 58, 58-60, 62-63, 72-73, 76-77, 93-94 and so on. Please double check the entire manuscript.
3. I do not understand the following sentence: Structural-based peptide design involves synthesizing peptides in vitro or engineering bacteria to produce peptides with specific structures. Please clarify this point. From my understanding, the peptide design is an approach or step commonly used before the chemical synthesis or recombinant expression of therapeutic peptides.
4. The quality of figure 1 should be improved.
5. The authors describe several techniques. They should include some examples of studies using these approaches to illustrate their potential and practical use. Is there some FDA-peptide or peptide in clinical trials designed using these technologies? If not, I also invite the authors to discuss the barriers or challenges for clinical translation.
6. I do not understand the reason for including solid phase approach in this manuscript. This is not an AI-based method. For this reason, I recommend delete this section.
7. There are several in silico tools for predicting the biological activity of peptides. I recommend the authors include them in this paper.
8.In the last section of the document, the authors discuss the role of reducing toxicity for therapeutic application, but do not present possible alternatives offered by in silico tools. Thus, I also recommend add information of AI tools freely available for studying the toxicity of peptides, particularly the haemolytic effect. For example, this manuscript (DOI: 10.3390/ph15030323) summarises a series of in silico approaches useful for design non-toxic peptides. These techniques should enrich this manuscript.
9. What are the possible applications of these therapeutic peptides? Clear examples are lacking for readers to understand the importance of advances in this area.
10. It would also be interesting to discuss the accuracy of these in silico models and approaches, including examples where the results were validated in vitro or in vivo. It is important to emphasize that the refinement and validation of the models is extremely important. In vitro and in vivo studies have also demonstrated failures of some models.
Author Response
Response to Reviewers’ Comments
We would like to start by thanking the anonymous reviewers for generously dedicating their time and offering insightful comments to enhance the quality of this manuscript.
Reviewer 1:
- The study is focused on peptide therapeutics. However sometimes the authors mentioned proteins or protein therapeutics. There are relevant differences between them. I suggest the authors does not change the focus of the study. My main recommendation is only describe peptide therapeutics. Please see lines 34, 46, 58 and so on.
Our response: Thank you for this suggestion. We have made the edits throughout the manuscript so that the focus remains on peptide therapeutics.
- Some sentences need to be supported by references. For instance, lines 45-50, 50-52, 58, 58-60, 62-63, 72-73, 76-77, 93-94 and so on. Please double check the entire manuscript.
Our response: We have updated the text accordingly at the indicated lines and throughout the manuscript.
- I do not understand the following sentence: Structural-based peptide design involves synthesizing peptides in vitro or engineering bacteria to produce peptides with specific structures. Please clarify this point. From my understanding, the peptide design is an approach or step commonly used before the chemical synthesis or recombinant expression of therapeutic peptides.
Our response: The confusing sentence is now modified.
- The quality of figure 1 should be improved.
Our response: We have updated figure one to better reflect the nature of the peptides and their interaction with a target protein.
- The authors describe several techniques. They should include some examples of studies using these approaches to illustrate their potential and practical use. Is there some FDA-peptide or peptide in clinical trials designed using these technologies? If not, I also invite the authors to discuss the barriers or challenges for clinical translation.
Our response: We have modified the manuscript to include numerous examples of peptides developed by each method. We have also included a new section entitled: “Barriers to FDA approval for computationally designed peptides”.
- I do not understand the reason for including solid phase approach in this manuscript. This is not an AI-based method. For this reason, I recommend delete this section.
Our response: We have removed this section of the manuscript. Similarly, we made major edits to the mimicry section of this manuscript (section 2). We noticed that certain parts of this section did not apply to peptide therapeutics. We have thus removed certain parts of this section.
- There are several in silico tools for predicting the biological activity of peptides. I recommend the authors include them in this paper.
Our response: We have modified the manuscript to include this information.
- In the last section of the document, the authors discuss the role of reducing toxicity for therapeutic application, but do not present possible alternatives offered by in silico tools. Thus, I also recommend add information of AI tools freely available for studying the toxicity of peptides, particularly the haemolytic effect. For example, this manuscript (DOI: 10.3390/ph15030323) summarises a series of in silico approaches useful for design non-toxic peptides. These techniques should enrich this manuscript.
Our response: We have modified the manuscript to include this information.
- What are the possible applications of these therapeutic peptides? Clear examples are lacking for readers to understand the importance of advances in this area.
Our response: We have modified the manuscript to include this information.
- It would also be interesting to discuss the accuracy of these in silico models and approaches, including examples where the results were validated in vitro or in vivo. It is important to emphasize that the refinement and validation of the models is extremely important. In vitro and in vivo studies have also demonstrated failures of some models.
Our response: We have highlighted the accuracy of various case studies, when available, relating to the model accuracy throughout the text. Also, we indicated the efficacy of the technologies by either reporting the number of peptides that were reported to display and effect, or any measures of accuracy as reported by the original authors. These points are included in the various case studies presented within the manuscript when directly indicated by the original authors.
In addition to these edits, we have also made several improvements to the grammar and syntax throughout the manuscript up to and including the title.

Reviewer 2 Report
Kazmirchuk et al reported a review about the current trend of computation and AI on effecting the peptide-protein interaction prediction. For sure, this topic is interested for the filed. However, the paper does not support enough case studies. Some tables to summarize the specific case studies will help audience to overview this topic. The relationship between peptide-protein interaction and structure prediction/ therapeutics need to be further clarified. More tools need to be included in this review, such as PepBind and Whey-derived peptides interaction, GalaxyPepDock, ClassAMP. In addition, the database for training will benefit for the paper, like DOMINO et al.
Author Response
Response to Reviewers’ Comments
We would like to start by thanking the anonymous reviewers for generously dedicating their time and offering insightful comments to enhance the quality of this manuscript.
Reviewer 2:
- Kazmirchuk et al reported a review about the current trend of computation and AI on effecting the peptide-protein interaction prediction. For sure, this topic is interested for the filed. However, the paper does not support enough case studies. Some tables to summarize the specific case studies will help audience to overview this topic. The relationship between peptide-protein interaction and structure prediction/ therapeutics need to be further clarified. More tools need to be included in this review, such as PepBind and Whey-derived peptides interaction, GalaxyPepDock, ClassAMP. In addition, the database for training will benefit for the paper, like DOMINO et al.
Our response: The text has now modified according to this comment. Additional examples of case studies are now included in the text.
In addition to these edits, we have also made several improvements to the grammar and syntax throughout the manuscript up to and including the title.
Round 2
Reviewer 1 Report
The new version of the manuscript has incorporated most of the reviewers' suggestions. The review is relevant and brings together important milestones in the area. However, small details escaped the authors.
1. Some sentences should be supported by references. For example:
These tools were developed due to synthetic peptides displaying various levels of toxicity.
This presents a major barrier to the large-scale adoption and application of peptides as therapeutics
In addition to their hemolytic activity, peptides can also display several specificity-related side effects such as the inappropriate association to immune receptors
Though tools are available to predict peptide toxicity through inappropriate interactions, peptides can still bind to other targets if not developed with an anti-selection
Generating computationally designed peptides for in-vivo applications is in its infancy.
A final barrier to large scale adoption of computationally designed peptides are manufacturing and cost-effectiveness.
2. Line 636. HemoPred does not predict immunogenic properties. Hence, please revise the following sentences.
Another peptide prediction tool is HemoPred. With the rapid and widespread development of synthetic peptide therapeutics, understanding their immunogenic properties is essential. H
Author Response
Response to Reviewers’ Comments
We would like to start by thanking the anonymous reviewers for generously dedicating their time and offering insightful comments to enhance the quality of this manuscript.
Reviewer 1:
- Some sentences should be supported by references. For example:
“These tools were developed due to synthetic peptides displaying various levels of toxicity.”
“This presents a major barrier to the large-scale adoption and application of peptides as therapeutics”
“In addition to their hemolytic activity, peptides can also display several specificity-related side effects such as the inappropriate association to immune receptors”
“Though tools are available to predict peptide toxicity through inappropriate interactions, peptides can still bind to other targets if not developed with an anti-selection”
“Generating computationally designed peptides for in-vivo applications is in its infancy.”
“A final barrier to large scale adoption of computationally designed peptides are manufacturing and cost-effectiveness.”
Our response: We have updated the text accordingly at the indicated passages.
- Line 636. HemoPred does not predict immunogenic properties. Hence, please revise the following sentences.
Another peptide prediction tool is HemoPred. With the rapid and widespread development of synthetic peptide therapeutics, understanding their immunogenic properties is essential. H
Our response: We have clarified the text accordingly at the indicated lines.
Reviewer 2 Report
The author addressed most of my comment. The font size in Figure 2 need to be larger for better visualization (e.g., SM1 et al).
Author Response
Response to Reviewers’ Comments
We would like to start by thanking the anonymous reviewers for generously dedicating their time and offering insightful comments to enhance the quality of this manuscript.
Reviewer 2:
- The author addressed most of my comment. The font size in Figure 2 need to be larger for better visualization (e.g., SM1 et al).
Our response: We have updated the font size in Figure 2.